# Exploring the Sustainability of China's New Energy Vehicle Development: Fresh Evidence from Population Symbiosis

Shengyuan Wang 

Business School, Nanjing Xiaozhuang University, Nanjing 211171, China; p2007007@njxzc.edu.cn

**Abstract:** It is particularly important to measure the growth prospects of new energy vehicles, especially electric vehicles, as they can effectively reduce the negative effects of the greenhouse effect. The population dynamics analysis model provides a method to comprehensively evaluate the growth mechanism, mode, and development prospects of new energy vehicles. In this research, the sales data of 20 automobile manufacturing enterprises were counted from the website database of the China Automobile Industry Association, and their development mechanism, development mode, and development trend were analyzed in order to help researchers understand the development prospects of China's new energy vehicle enterprises. The conclusion is that the analysis results of the single population logistic model show that the intrinsic growth rate of Chinese new energy vehicle enterprises is generally relatively low. The intrinsic growth rate of China's new energy automobile enterprises is lower than that of other mature traditional automobile manufacturing enterprises in China. The level of intrinsic growth rate of new energy vehicle enterprises is similar to that of declining enterprises with significantly declining sales. The Lotka–Volterra model provides the analysis results of the growth mechanism driven by market demand of automobile manufacturing sample enterprises. The market driven mode of China's new energy vehicle enterprises is not obvious. It is difficult for the current development mechanism of China's new energy vehicle enterprises to achieve the sustainability of growth. The optimization results of the MCGP model show that China's new energy vehicle enterprises should transform to a market-driven development model.

**Keywords:** new energy vehicle; population symbiosis; Lotka–Volterra model; logistic model

## 1. Introduction

With the rapid development of economy, the human industrial economy has made great achievements, but the damage to the environment is also increasing, especially non-renewable energy is facing the depletion crisis [1]. The energy crisis has become an important constraint to economic development. Moreover, the traditional automobile industry produces a lot of harmful pollutants as it consumes a lot of petroleum energy. Environmental pollution and global warming caused by the massive use of petrochemical energy have also become a common concern worldwide. At the beginning of the 21st century, with the rapid development of science and technology, especially the proposal of sustainable development strategy, the new energy vehicle industry, represented by hybrid electric vehicles, hydrogen-powered vehicles, pure electric vehicles, and fuel cell vehicles, has achieved rapid development with the strong support of governments around the world [2]. Under the dual urgent situation of energy crisis and environmental pollution, the transition of automobile industry to new energy has become the common strategic choice of all countries in the world, and its promotion and application in the world has also become a general trend. The development of new energy vehicles has important strategic and practical significance in breaking through the energy bottleneck, meeting the technological upgrading, and embodying the concept of low-carbon development [3].

As a capital-, technology-, and talent-intensive industry, the automobile industry plays an important role in the national economy, and it is the main driving force for the

development of the national economy. It is highly correlated and plays a great role in driving-related industries, especially as the fastest-growing industry in the future; whether it can achieve greater development is directly related to the growth rate of the national economy. However, all countries in the world are vigorously promoting the development of new energy vehicle industry, resulting in the faster and faster application of new technologies [4]. It is now an urgent problem to be solved concerning how to speed up the development of China's new energy vehicle industry to cope with the fierce competition in the world's new energy vehicle industry, seize the new highland of industrial technology development, and win the competitive advantage [5]. Therefore, it is undoubtedly of great theoretical and practical significance to speed up the development of China's new energy vehicle industry to empirically analyze the law of technology collaborative diffusion of China's new energy vehicle industry, find out the factors that restrict the technology collaborative diffusion of China's new energy vehicle industry, analyze the promotion path of new technology application diffusion, have an insight into the power source of the technology collaborative diffusion system of the new energy vehicle industry, and formulate reasonable and effective promotion measures [6]. "Rules for New Energy Vehicle Production Enterprises and Product Access Management" formulated in July 2009 by the Chinese government clearly defined new energy vehicles, emphasizing that new energy vehicles refer to vehicles with advanced principles and new structures that use unconventional vehicle fuels as power and use newly developed drive and power control technologies in the automotive industry. In the 21st century, the key for China to realize the transformation from a large automobile manufacturing country to a powerful automobile manufacturing country lies in the development of new energy vehicles, the formation of a complete industrial system and innovation system from key parts to complete vehicles, and the synchronization of the overall technical level with the international advanced level. New energy vehicles mainly include hybrid electric vehicles, pure electric vehicles, fuel cell vehicles, etc. Compared with traditional vehicles, they have the following characteristics: technological innovation, product efficiency, and industrial dependence.

At present, most emerging economies in the world have entered a new stage of development, and cars are gradually popularized in these countries, which will be an unbearable test for the environment. Global warming caused by the consumption of fossil fuels will also have an impact on the physical flow of global trade, mainly reflected in the refrigeration of goods and the impact on engine efficiency [7]. In order to alleviate the above-mentioned energy crisis and environmental problems, people place their hopes on the development and use of new energy. The fossil fuel consumed by automobiles is an important component of global energy consumption, and it is also the main emission source of greenhouse gases such as carbon dioxide. Scientists naturally focus on the application and promotion of new energy vehicles.

People's concerns about global warming, urban air pollution, and dependence on unstable and expensive foreign oil supplies have prompted policymakers and researchers to study new energy vehicles, an alternative to conventional petroleum fueled internal combustion engine vehicles in transportation [8]. Because vehicles that obtain part or all of the power from the electric drive system can have low-temperature or even zero emissions of greenhouse gases and urban air pollutants, and can consume little or no oil, the development and research of advanced electric vehicles have considerable incentives. They quantified the benefits of vigorously promoting new energy vehicles for researchers and policymakers from the aspects of new energy vehicle technology, supporting infrastructure required for development, carbon emissions in the operation of new energy vehicles, consumer acceptance, performance parameters of new energy vehicles, etc.

Environmental factors are one of the main reasons for the Chinese government to vigorously develop new energy vehicles. The rapid development of China's photovoltaic industry in the past few years has also laid the foundation for the development of China's new energy vehicles [9].

In China, wind energy and other renewable energy also have an impact on the new energy vehicle industry. However, the development of wind energy and other renewable energy has just started. There are few relevant data and studies. At the same time, the construction of supporting facilities for new energy vehicles also has an important impact on the development of the new energy vehicle industry, for example, whether people are willing to support the construction of charging facilities [10]. The construction of intelligent charging facilities is an important constraint for the development of new energy vehicles that scholars are generally concerned about [11–13]. With the development of China's new energy vehicle industry, issues such as production and sales prospects [14] and supply chain risks [15] have also entered the research field of scholars.

As an important way to alleviate the energy crisis and environmental crisis, new energy vehicles have been widely recognized by researchers. For China, the development of this industry is also closely related to the transformation and upgrading of the entire economy. Based on the concept of sustainable ecological development, only by forming its own evolution mechanism and development mode and building a good innovation ecosystem can the new energy automobile industry break through industrial barriers, occupy the commanding heights of the core technology of the automobile industry, and obtain sustainable industrial competitive advantages.

It is of great significance to quantitatively evaluate the development level of new energy vehicles and analyze the effect of supporting policies in this field. Scholars mostly adopt a combination of subjective and objective methods to study the development level of new energy vehicles [16,17]. Comprehensive evaluation methods such as entropy weight analytic hierarchy process [18] and fuzzy set analytic hierarchy process [19] are used to analyze the development level of new energy vehicles. At present, the development of new energy vehicles mainly depends on government policy incentives and market drivers. Scholars pay attention to the market performance of new energy vehicles, such as evaluating consumers' willingness to use new energy vehicles [20,21]. Policy research on the development of new energy vehicles has also been prevalent [22].

China's goal of promoting technological progress is effective in developing the local new energy vehicle industry [3], but it is still unclear whether China will succeed in "surpassing the curve" [23]. The development status of new energy vehicle patents does not indicate that China has a leading edge in new energy vehicle technology. China's new energy vehicle industry policy should be further strengthened, especially the core policy of technological innovation [24]. Relevant research shows that China's new energy vehicle industry has changed from "government-driven" to "government-driven + market-driven" [25]. When the market and academia formed a research trend on the development of new energy vehicles, scholars also began to explore the actual technical effect of new energy vehicles in the field of emission reduction. From the perspective of the effect of environmental protection and new energy technology [26], new energy vehicles are close to zero-emissions vehicles, which are of great significance to achieve the coordinated development of environmental, social, and health goals [27]. The zero-emission of new energy vehicles is very suitable for alleviating air pollution [28]. However, during the operation of new energy vehicles, the burden of driving emissions is transferred to power plants. The power generation configuration of an economy has greatly affected the environmental improvement efficiency of new energy vehicles in relevant regions [29–32]. New energy vehicles will still increase environmental pollution, such as particle formation [33,34].

Through the research and mining of previous research results, the author finds that (1) the research on new energy vehicles mainly focuses on the industrial development and the constraints of technology and supply chain, and few articles focus on the management of new energy vehicle enterprises. Therefore, this paper intends to carry out research on new energy vehicles from the perspective of the development of new energy vehicle enterprises. (2) The existing research focuses on technical indicators and problems such as carbon emissions and power batteries of new energy vehicles, and lacks research on the overall symbiosis between new energy vehicles and the automobile industry. This

issue involves industrial development and competition and cooperation of enterprise brands. (3) Previous scholars have more static analysis and less dynamic consideration in quantitative research, which greatly weakens the persuasiveness of the conclusion. Most evaluations are based on the construction of evaluation index system, and there is less research on the dynamic mechanism of enterprise development.

Based on the in-depth research and analysis of the theory of innovation ecosystem, this paper attempts to build a theoretical model of population dynamics for the development of China's new energy vehicle industry. On this basis, the sustainable development level of China's new energy vehicle industry is measured. The MCGP model is built to analyze the evolution direction of the sustainable development of China's new energy vehicle industry. The research results are expected to enrich and improve the research system of socio-economic ecosystem theory and provide a new perspective for the development of China's new energy vehicle industry.

## 2. Methodology and Data

The research idea of this paper is shown in the following Figure 1:

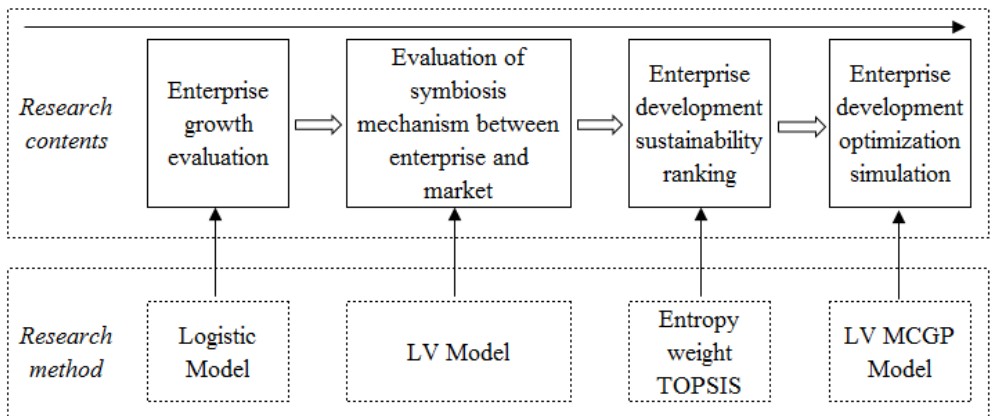

**Figure 1.** Research design of this paper.

As shown in Figure 1, the paper extends the logistic and LV models to the research on the growth mechanism of NEV, and incorporates the model estimation parameters into the TOPSIS evaluation system, and then uses the MCGP model to analyze the optimization path of the growth model of new energy vehicle companies.

### 2.1. Socio-Economic Ecosystem and Population Dynamics

The main idea of this paper is to regard the automobile sales volume of sample enterprises as the population of automobile products, and to view the growth of enterprises with the characteristics of population growth. Based on the two-species LV model, the symbiotic relationship between the sales volume of the enterprise and the total sales volume of the automobile market is analyzed. The application of population dynamics to the analysis of social and economic systems has a long history. This section briefly combs the theoretical development process of socio-economic ecosystem.

Ecosystem is an economic community supported by interacting "organizations and individuals" [35]. Business ecosystem research focuses on the concept of coevolution [36–38]. The concept of ecosystem is only used to describe a certain system, and there is little in-depth analysis of the system operation mechanism based on systematic insights [39]. The hierarchical structure of ecosystem helps to control the activities of producing goods. Promoting coevolution among populations is a basic function of ecosystems. The basis of coevolution should be understood as an opportunity space, which is a relatively unbounded, open, and unexplored region. An ecosystem is affected by both the downward force of relatively stable institutions at the macro level and the upward force between

interactive actors at the micro level [40,41]. Coevolutionary dynamics can occur not only within ecosystems, but also between ecosystems and their external environments [42,43].

In organizational research, multi-level characteristics are the core attributes of coevolution [44]. In the empirical study of ecosystem, both micro and macro methods have been proven to be effective [45,46]. From the perspective of macro coevolution, ecosystem and environment carry out uninterrupted exchange of material, knowledge, and information to achieve sustainable innovation. In order to cope with the emerging opportunities and threats, the ecosystem is dynamic and open, further increasing the interdependence, vitality, and instability between different populations in the ecosystem [47]. The existing business ecosystem research can help us better understand the driving force of single innovation, the continuous innovation of single business, or the continuous innovation of corporate business.

Coevolution is essentially oriented by continuous innovation. The only way for a company to fight against continuous competition and commercialization is to become a continuous innovator. The coevolutionary ecosystem includes the combination of markets and enterprises, including new and existing markets, as well as new and existing enterprises. The view of coevolution assumes that the basic task of ecosystem is to promote sustainable innovation and regards ecosystem as a complex adaptive system whose function is to resist external shocks and take advantage of external opportunities. In the absence of a clear role of the external environment, taking a given value proposition as the boundary of an endogenous ecosystem is actually defining the concept of ecosystem as a semi-closed system [48]. The mechanism of coevolution lies in the benign evolution of symbiotic relationships between populations.

The development of any population will be restricted by its own growth capacity, resources, and environment. Almost all species obey the law of life cycle. Similarly, the development of population in the socio-economic system should not be unlimited growth. If a socio-economic system is regarded as an ecosystem, the species in the socio-economic system can be regarded as populations in the ecosystem. The population dynamics model mainly focuses on the change of population number, and its change law is based on the nonlinear growth law of biological population number. Many species in nature grow nonlinearly, which is a very common phenomenon. The competition and coordination mechanism within the population is also an important factor. This setting is based on the principle of intraspecific competition of biological populations. There is competition among natural biological populations. The more species there are, the fiercer the competition will be. Therefore, this mechanism should also become an important part of the population growth model.

Logistic function was first proposed by Verhulst, a Belgian scholar [49]. Cox, a British statistician, proposed a logistic regression model in 1958 [50]. The advantage of this model is that there are not many requirements in normality, homogeneity of variance, and the interpretability of coefficients, which makes logistic regression model widely used in many fields such as medicine and social investigation.

Logistic regression model has been widely used in the past for many years. For example, it has been used in the research of infectious diseases from the beginning. As an effective data processing method, logistic regression analysis is widely used in biology and ecological engineering, medicine, criminology, management, economics, sociology, linguistics, pedagogy, and other fields. The logistic regression model has achieved similar results in statistics. According to the logistic model, the growth dynamic system within population 1 (P1) is constructed.

$$g_1(t) = \frac{dN_1(t)}{dt} = \alpha_1 N_1 \left(1 - \frac{N_1}{K_1}\right) \tag{1}$$

$g_1(t)$ is the growth rate of stage T.
$N_1(t)$ is the population size of T period.
$K_1$ is the largest population size.

$\alpha_1$ is the intrinsic growth rate.

$\left(1 - \frac{N_1}{K_1}\right)$ is growth retardation factor.

The measurement model is as follows:

Because: $dN_1(t) \approx \Delta N_1(t)$, $\Delta N_1(t) = N_1(t) - N_1(t-1)$, $dt \approx \Delta t = t - (t-1) = 1$.

Therefore:

$$g_1(t) \approx \Delta N(t) = \gamma_1 N_1(t-1) + \gamma_2 N_1^2(t-1) \tag{2}$$

Set $\gamma_1 = \alpha_1$. Normally, $\gamma_1 > 0$, representing the synergy within a group, and it is called internal synergy coefficient. When $\gamma_1 > 1$, there are significant synergistic effects in the population.

Set $\gamma_2 = -\frac{\alpha_1}{K_1}$. Normally, $\gamma_2 < 0$, and is used to express the competition effect within the population, which is called the internal competition coefficient or the population density inhibition coefficient.

Lotka–Volterra (LV) system. Based on the logistic model of a single species, the LV model takes into account the dynamic growth of simultaneous competition and symbiosis between two or more entities in the ecosystem [51] and can accurately describe the competition and symbiosis between enterprise groups. The LV system can determine the influence of core population in the evolution of the whole ecosystem [52], so it has better data fitting and prediction expression [53].

$$\begin{cases} g_1(t) = \frac{dN_1(t)}{dt} = \alpha_1 N_1\left(1 - \frac{N_1}{K_1} + \frac{\beta_{12} N_2}{K_2}\right) \\ g_2(t) = \frac{dN_2(t)}{dt} = \alpha_2 N_2\left(1 - \frac{N_2}{K_2} + \frac{\beta_{21} N_1}{K_1}\right) \end{cases} \tag{3}$$

The classical Lotka–Volterra model is a differential dynamic system, which is used to simulate the dynamic relationship between populations in ecology. Later, economists introduced it into the fluctuation of macroeconomic growth and medium-sized and wide-ranging market competition. There is also a relationship between market competition subjects: the existence of competition subjects can promote or inhibit the diffusion process of another competition subject. The interaction type between species can be judged based on the value [54].

## 2.2. Evaluation of Enterprise Development Sustainability Based on Entropy Weight TOPSIS

The entropy weight method and TOPSIS evaluation method have been successfully applied in the field of environment and sustainable development [55,56], and the author has combined them to study the evaluation of urbanization and water resource efficiency [57]. The practice shows that this method is suitable for evaluating the research indicators of this paper.

Based on the similarity between ideal solution and observation data, the development sustainability of different enterprises is evaluated. The higher the similarity, the better the growth. Based on the similarity between ideal solution and observation data, the development sustainability of different enterprises is evaluated. The higher the similarity, the better the development. The evaluation matrix is $A$.

$$A = \left[a_{ij}\right]_{m \times n} \tag{4}$$

In this study, the entropy weight method is used to determine the weight of logistic and LV model parameters. The Technique for Order Preference by Similarity to Ideal Solution (TOPSIS) is used to evaluate the similarity. The ideal scale (maximum value) of different enterprises can be used as the evaluation standard. The entropy weight method is provided as follows. Suppose m is the number of enterprises ($A_1, A_2, \ldots, A_m$), and $N$ is the development sustainability evaluation parameter ($C_1, C_2, \ldots, C_n$). Among them: ai1 = $\alpha$1 (Logistic Model), ai2 = $\gamma$2 (Logistic Model), ai3 = K1 (Logistic Model), ai4 = $\alpha$1(LV Model), ai5 = $\gamma$2 (LV Model), ai6 = K1 (LV Model), ai7 = $\gamma$3 (LV Model), ai8 = $\beta$12 (LV Model).

Then, the initial evaluation matrix is:

$$A = \begin{bmatrix} a_{11} & a_{12} & \cdots & a_{1n} \\ a_{21} & a_{22} & \cdots & a_{2n} \\ \vdots & \vdots & \ddots & \vdots \\ a_{m1} & a_{m2} & \cdots & a_{mn} \end{bmatrix} = \left[ a_{ij} \right]_{m \times n} \tag{5}$$

Step 1: standardize the evaluation matrix

$$r_{ij} = \frac{a_{ij}}{\sqrt{\sum_{i=1}^{m} a_{ij}^2}} \tag{6}$$

Step 2: calculate entropy

$$e_j = -\frac{1}{\ln m} \sum_{i=1}^{m} r_{ij} \ln r_{ij}, \, j = 1, 2, \cdots, n \tag{7}$$

Step 3: calculate weights

$$w_j = \frac{1 - e_j}{\sum_{i=1}^{n} (1 - e_j)}, j = 1, 2, \cdots, n \tag{8}$$

The TOPSIS method is shown below.

Step 1: construct normalized matrix R

$$r_{ij} = \frac{a_{ij}}{\sqrt{\sum_{i=1}^{m} a_{ij}^2}} \tag{9}$$

Step 2: construct the weighted normalization matrix V

$$v_{ij} = w_j r_{ij}, \sum_{j=1}^{n} w_j = 1, \tag{10}$$

$w_j$ is the weight of the $j$th standard.

Step 3: calculate A$^+$ and A$^-$

A$^+$ and A$^-$ are defined as follows.

$$A^+ = \left\{ (\max v_{ij} | j \in J) \, or \, (\min v_{ij} | j \in J\prime) \right\}, i = 1, 2, \cdots, m \\ = \left\{ v_1^+, v_2^+, \cdots, v_n^+ \right\} \tag{11}$$

$$A^- = \left\{ (\min v_{ij} | j \in J) \, or \, (\max v_{ij} | j \in J\prime) \right\}, i = 1, 2, \cdots, m \\ = \left\{ v_1^-, v_2^-, \cdots, v_n^- \right\} \tag{12}$$

The positive ideal solution (PIS) and negative ideal solution (NIS) are determined. $J$ and $j$ are positive and negative standard sets, respectively.

Step 4: calculate the distance between each enterprise evaluation data and positive ideal standard set (PIS) and negative ideal standard set (NIS)

$$S_i^+ = \sqrt{\sum_{j=1}^{n} \left( v_{ij} - v_j^+ \right)^2}, i = 1, 2, \cdots, m. \tag{13}$$

$$S_i^- = \sqrt{\sum_{j=1}^{n} \left( v_{ij} - v_j^- \right)^2}, i = 1, 2, \cdots, m. \tag{14}$$

Step 5: sort the order of sustainable development of enterprises

$$C_i^+ = \frac{S_i^-}{S_i^+ + S_i^-}, 0 < C_i^+ < 1, i = 1, 2, \cdots, m. \tag{15}$$

Here is $C_i^+ \in [0, 1]$, where $i = 1,2,\ldots, M$. Therefore, the best enterprise should be found in the order of $C_i^+$. The larger is the value of $C_i^+$, the better. If $C_i^+$ is close to 1, the alternative $A_i$ is closer to PIS.

### 2.3. Lotka–Volterra MCGP Model

This paper constructs a Lotka–Volterra MCGP model for the optimization data simulation of the development of new energy vehicle manufacturing enterprises under the market driven mode. Based on the progress of relevant models [58–60], this problem can be expressed as follows:

$$\begin{cases} Min \ d_1^+ + d_1^- + e_1^+ + e_1^-; \\ g_1(t) = \frac{dN_1(t)}{dt} = \alpha_1 N_1 \left(1 - \frac{N_1}{K_1} + \frac{\beta_{12} N_2}{K_2}\right); \\ g_2(t) = \frac{dN_2(t)}{dt} = \alpha_2 N_2 \left(1 - \frac{N_2}{K_2} + \frac{\beta_{21} N_1}{K_1}\right); \\ -1 < \beta_{12} < 1; -1 < \beta_{21} < 1; \\ N_1 = k_1 + d_1^+ + d_1^-; N_2 = k_2 + e_1^+ + e_1^-; \\ d_1^+ >= 0; d_1^- >= 0; e_1^+ >= 0; e_1^- >= 0. \end{cases} \tag{16}$$

The research object of this study meets the requirements of ecosystem. The data of non-symbiotic system are not suitable for analysis with this model. Relevant studies have verified the stability of the model, and different types of data will not affect the use of the model [58–60].

### 2.4. Sample Selection

As the New Energy Vehicle (NEV) enterprise is a relatively new concept, there is no specific industry division standard. Some automobile manufacturing enterprises also began to produce new energy vehicles while producing fuel vehicles. The sample enterprises studied in this paper are those mainly engaged in the production of new energy vehicles. This study is based on the data of "average fuel consumption and new energy vehicle points of Chinese passenger car enterprises". Enterprises with high credits have been selected as the main research sample. On 5 July 2022, the Ministry of Industry and Information Gechnology of China issued an announcement on the average fuel consumption and new energy vehicle credit of Chinese passenger vehicle enterprises in 2021 ("Dual-Credit Policy"). Overall, the performance of self-owned brand auto enterprises is better than that of joint venture auto enterprises. Among them, SAIC GM Wuling Automobile Co., Ltd. and Tesla (Shanghai, China) Co., Ltd. ranked first in the average fuel consumption ranking and new energy vehicle ranking, respectively; SAIC General Motors Co., Ltd. and Dongfeng Motor Co., Ltd. ranked last in the average fuel consumption ranking and new energy vehicle ranking, respectively. In the "Dual-Credit Policy" list in 2021, companies with strong new energy vehicle products, such as BYD and Tesla, became the "big players" of credit. It is worth mentioning that the new forces of vehicle manufacturing, which mainly focus on new energy vehicles, have generally performed well. The new energy vehicle credits of Xiaopeng automobile, Hozon automobile, WM automobile, ideal automobile, and Leap automobile are 360,900, 208,400, 159,300, 143,500, and 82,600 respectively. (Data source: https://www.miit.gov.cn/jgsj/zbys/qcgy/art/2022/art_031bc64d8eaf4064a31f6 6d714603438.html, accessed on 6 July 2022).

In general, the integral performance is relatively good with high sales volume of new energy vehicles. In fact, the number of new energy vehicles in China has exceeded 7.8 million since the implementation of the "double points" policy in the automotive industry five years ago. An important purpose of China's implementation of "double points" is to increase the output of energy-saving vehicles and new energy vehicles, which plays a significant role in promoting energy conservation and new energy vehicle development, transportation, and other fields in China. This paper selects the new energy vehicle manufacturing enterprises with more than 80,000 points as the sample. The samples include BYD, Tesla (Shanghai, China), Xiaopeng automobile, Hozon automobile, WM automobile,

Lixiang automobile, Nio automobile, and Leap automobile. Based on the modeling needs of population dynamics model, this paper mainly selects the sales volume index of cars as the main index of modeling. In this paper, the sample data are analyzed by single population growth analysis based on logistic model, market driven growth trend analysis based on two-dimensional LV model, and growth mode optimization path analysis based on LV MCGP model.

## 3. Empirical Analysis

### 3.1. Growth Analysis of Single Population Based on Logistic Model

In this paper, we first analyze the growth of single population based on the logistic model. The sales volume data of relevant enterprises are shown in Table 1.

**Table 1.** Monthly sales volume of China's new energy vehicle sample enterprises (unit: vehicle).

| Report Date | BYD | Tesla China | Xiaopeng | Hozon | WM | Lixiang | Nio | Leap |
|---|---|---|---|---|---|---|---|---|
| May-22 | 114,183 | 32,165 | 10,125 | 11,009 | 3003 | 11,496 | 7024 | 10,069 |
| Apr-22 | 105,475 | 1512 | 9002 | 8813 | 1521 | 4167 | 5074 | 9087 |
| Mar-22 | 103,852 | 65,814 | 15,414 | 12,026 | 3719 | 11,034 | 9985 | 10,059 |
| Feb-22 | 88,093 | 56,515 | 6225 | 7117 | 1557 | 8414 | 6131 | 3435 |
| Jan-22 | 93,363 | 59,845 | 12,922 | 11,009 | 2200 | 12,268 | 9652 | 8085 |
| Dec-21 | 97,990 | 70,847 | 16,000 | 10,127 | 5062 | 14,087 | 10,352 | 7046 |
| Nov-21 | 97,242 | 52,859 | 15,613 | 10,013 | 5027 | 13,485 | 10,400 | 5775 |
| Oct-21 | 88,898 | 54,391 | 10,138 | 8107 | 5025 | 7649 | 5225 | 3827 |
| Sep-21 | 79,037 | 56,006 | 10,168 | 7699 | 2635 | 7094 | 9227 | 3766 |
| Aug-21 | 62,848 | 44,264 | 7265 | 6613 | 3627 | 9433 | 4365 | 4270 |
| Jul-21 | 56,975 | 32,968 | 7460 | 6011 | 4027 | 8589 | 8800 | 4157 |
| Jun-21 | 49,765 | 33,155 | 7061 | 5138 | 4007 | 7713 | 8438 | 4050 |
| May-21 | 45,176 | 33,463 | 5944 | 4508 | 3082 | 4323 | 6822 | 3121 |
| Apr-21 | 44,606 | 25,845 | 5605 | 4015 | 3027 | 5539 | 8155 | 2864 |
| Mar-21 | 37,189 | 35,478 | 4423 | 3246 | 2503 | 4900 | 7449 | 2863 |
| Feb-21 | 19,529 | 18,318 | 3035 | 2002 | 1006 | 2300 | 5890 | 393 |
| Jan-21 | 42,094 | 15,484 | 5180 | 2195 | 2040 | 5379 | 7748 | 1668 |
| Dec-20 | 55,075 | 23,804 | 6420 | 3015 | 2588 | 6126 | 6623 | 3024 |
| Nov-20 | 52,806 | 21,604 | 4650 | 2122 | 3018 | 4646 | 5500 | 2032 |
| Oct-20 | 46,560 | 12,143 | 815 | 2056 | 3003 | 3692 | 5145 | 1743 |
| Sep-20 | 40,905 | 11,329 | 853 | 2023 | 2107 | 3504 | 5003 | 1050 |
| Aug-20 | 30,024 | 11,811 | 623 | 1205 | 2057 | 2711 | 3761 | 928 |
| Jul-20 | 27,890 | 11,014 | 551 | 1016 | 2036 | 2445 | 3680 | 884 |
| Jun-20 | 31,738 | 14,954 | 821 | 1333 | 2028 | 1834 | 4018 | 879 |

As shown in Table 1, the product sales volume of China's new energy vehicle manufacturing enterprises varies greatly. BYD and Tesla China rank among the top in the sales of new energy vehicles, and the sales of BYD are several times that of Tesla China in each observation period. The monthly sales volume of several other new energy vehicle enterprises is about 10,000. It can be seen that BYD is the core and leading enterprise in the field of new energy vehicles in China. The change trend of sales data of relevant new energy sample enterprises is shown in Figure 2.

As shown in Figure 2, there are significant differences in the sales trend of new energy vehicles of different enterprises. BYD and Tesla China have similar sales growth trends, and the total sales volume of other automobile enterprises has been hovering at a relatively low level. The sales data of Tesla China fell precipitously in April 2022. The main reason was that the epidemic prevention and control measures in Shanghai at that time led to the production stagnation of Tesla Shanghai factory.

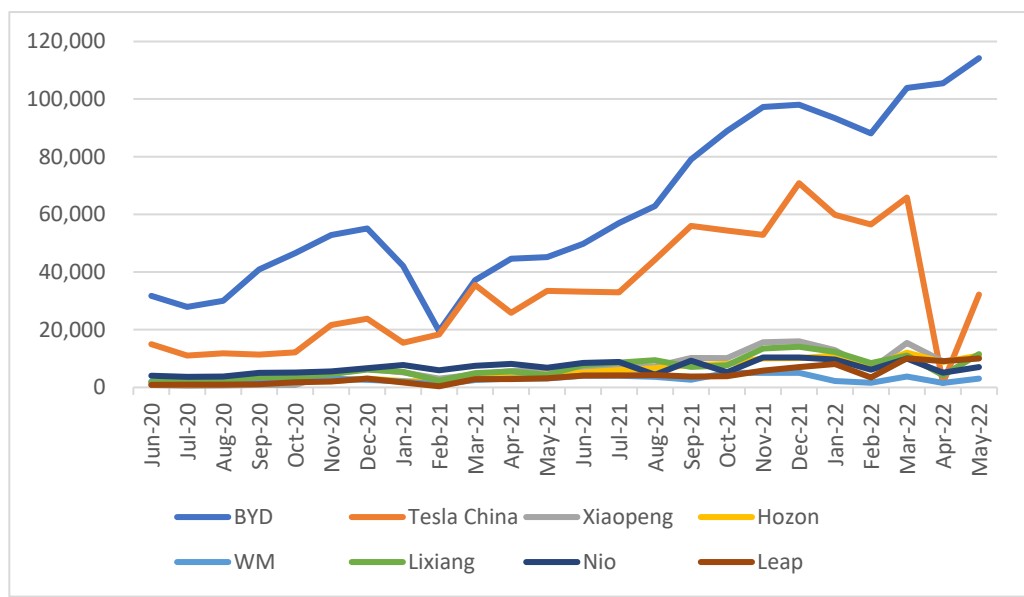

**Figure 2.** Sales trend of new energy vehicles.

As shown in Table 2, the regression effect of the model is very good, and the logistic regression of most enterprise sales data can pass the significance test. The intrinsic growth rate of all new energy vehicle manufacturing sample enterprises is at a relatively low level. In the theory of population dynamics, when the intrinsic growth rate is greater than 1, it can be said that the intrinsic growth attribute of the population is better. None of the sample enterprises has an intrinsic growth rate greater than 1. The best intrinsic growth rate is 0.919 of Nio automobile, and the intrinsic growth rate of most enterprises is about 0.5. BYD's intrinsic growth rate was 0.104, the lowest among the sample enterprises. In order to better analyze the growth trend of automobile manufacturing enterprises, this paper introduces the fuel vehicle enterprises with the highest sales volume and the enterprises with declining sales volume as a comparative sample.

As shown in Table 3, this paper selects 8 enterprises that rank among the top in the national total automobile sales and fuel vehicle sales as the comparison sample. Among these enterprises are famous Sino foreign joint venture automobile enterprises, such as FAW VW, GAC Toyota, BMW Brilliance, etc. There are also self-owned brand enterprises, such as Chang'an, Geely, etc. During the sample observation period, the total number of cars sold in China can reach 2,398,523 when it is high, and around 1 million when the sales are low. The lowest sales volume in Table 3 appeared in April 2022, which was mainly due to the impact of the epidemic and prevention and control measures. Figure 3 shows the change trend of the data of enterprises with the highest sales data in China's automobile market.

As shown in Figure 3, the change trend of monthly sales data of fuel vehicle enterprises with the highest sales volume shows a significant oscillation. The volatility of sales data shows the following characteristics: (1) the sales situation of traditional automobile sales enterprises is affected by many factors, and there are many market interference factors. (2) China's automobile market has not reached an equilibrium state, and the automobile sales are in a state of oscillation. The mismatch between supply and demand leads to market fluctuations. (3) The sales volume in the observation period has an oscillatory downward trend, which is mainly caused by the background of the development of China's automobile market.



**Table 2.** Single population growth analysis of China's new energy vehicles based on logistic model.

| Enterprise | Intrinsic Growth Rate ($\alpha1$) | Internal Inhibition Coefficient ($\gamma2$) | Theoretical Upper Limit of Sales Volume (K1) |
|---|---|---|---|
| BYD | 0.104 (0.945) | −0.00000068 (−0.524) | 153,407 |
| Tesla China | 0.550 (1.790) * | −0.00001201 (−2.141) ** | 45,795 |
| Xiaopeng | 0.563 (2.393) ** | −0.00005074 (−2.729) *** | 11,098 |
| Hozon | 0.577 (3.443) *** | −0.00006184 (−3.406) *** | 9327 |
| WM | 0.487 (1.866) * | −0.00014673 (−2.198) ** | 3317 |
| Lixiang | 0.471 (1.907) * | −0.00004985 (−2.133) ** | 9452 |
| Nio | 0.919 (3.630) *** | −0.00011973 (−3.947) *** | 7680 |
| Leap | 0.390 (1.552) * | −0.00005330 (−1.593) * | 7320 |

*() t* value, * *p* value < 0.1, ** *p* value < 0.05, *** *p* value < 0.01.

**Table 3.** Monthly sales volume of China's leading sample enterprises in automobile sales.

| Report Date | Total National Sales | Faw-VW | Gac-Toyota | Saic-VW | SGMW | Chang'an | Geely | BMW-Brilliance | Dongfeng-Nissan |
|---|---|---|---|---|---|---|---|---|---|
| May-22 | 1,576,803 | 89,025 | 83,730 | 83,502 | 71,493 | 66,091 | 60,197 | 62,567 | 52,531 |
| Apr-22 | 950,343 | 39,444 | 68,450 | 28,685 | 44,002 | 47,980 | 49,137 | 31,743 | 37,636 |
| Mar-22 | 1,819,405 | 76,586 | 96,984 | 104,200 | 102,951 | 110,015 | 75,447 | 35,723 | 56,114 |
| Feb-22 | 1,451,420 | 70,638 | 49,710 | 86,076 | 43,645 | 53,034 | 55,357 | 43,558 | 74,308 |
| Jan-22 | 2,138,181 | 103,462 | 99,707 | 124,491 | 72,639 | 123,707 | 112,325 | 79,087 | 110,996 |
| Dec-21 | 2,398,523 | 113,635 | 93,587 | 130,878 | 151,144 | 64,830 | 122,056 | 51,427 | 88,326 |
| Nov-21 | 2,175,564 | 87,518 | 81,099 | 127,201 | 128,951 | 76,113 | 103,497 | 47,158 | 92,360 |
| Oct-21 | 1,990,339 | 85,096 | 56,921 | 112,400 | 115,808 | 82,402 | 86,047 | 54,836 | 78,971 |
| Sep-21 | 1,737,510 | 58,593 | 44,704 | 116,840 | 75,343 | 72,032 | 84,500 | 53,837 | 74,297 |
| Aug-21 | 1,543,903 | 57,844 | 38,756 | 117,644 | 100,033 | 62,997 | 77,278 | 58,511 | 80,662 |
| Jul-21 | 1,543,474 | 39,391 | 75,130 | 68,451 | 72,446 | 70,200 | 79,185 | 43,466 | 74,813 |
| Jun-21 | 1,553,528 | 53,688 | 73,210 | 63,671 | 61,571 | 68,086 | 81,502 | 59,640 | 77,078 |
| May-21 | 1,642,018 | 96,495 | 70,018 | 107,370 | 69,914 | 75,820 | 76,575 | 62,858 | 73,864 |
| Apr-21 | 1,746,754 | 67,003 | 73,900 | 101,349 | 79,732 | 83,912 | 80,549 | 61,303 | 79,744 |
| Mar-21 | 1,914,414 | 129,871 | 68,800 | 107,537 | 82,734 | 83,737 | 82,668 | 65,543 | 72,746 |
| Feb-21 | 1,148,130 | 69,160 | 41,500 | 48,039 | 40,957 | 81,934 | 64,860 | 41,696 | 50,985 |
| Jan-21 | 2,358,372 | 120,848 | 89,800 | 85,422 | 60,933 | 114,048 | 129,644 | 73,333 | 108,274 |
| Dec-20 | 2,285,751 | 123,029 | 72,159 | 145,983 | 112,855 | 68,887 | 127,932 | 54,834 | 121,886 |
| Nov-20 | 2,098,448 | 154,391 | 77,400 | 145,735 | 95,663 | 97,054 | 125,712 | 61,219 | 117,430 |
| Oct-20 | 2,300,447 | 141,050 | 72,000 | 137,300 | 84,716 | 95,266 | 116,244 | 47,166 | 110,507 |
| Sep-20 | 2,075,889 | 137,077 | 81,000 | 156,839 | 75,526 | 81,796 | 102,451 | 56,350 | 110,523 |
| Aug-20 | 1,754,600 | 112,508 | 66,314 | 129,046 | 64,770 | 73,831 | 91,641 | 65,558 | 101,901 |
| Jul-20 | 1,664,826 | 92,150 | 73,952 | 122,000 | 50,506 | 69,657 | 86,508 | 63,596 | 93,787 |
| Jun-20 | 1,720,593 | 105,421 | 66,888 | 127,794 | 43,151 | 68,608 | 92,593 | 46,597 | 106,570 |

As shown in Table 4, the regression effect of the model is very good, and the logistic regression of most enterprise sales data can pass the significance test. The intrinsic growth rate of automobile manufacturing sample enterprises is uneven. The intrinsic growth rate of China's total sales is 0.697, which shows that the growth momentum of China's auto sales is relatively stable. Among the sample enterprises, the intrinsic growth rates of Chang'an, Gac-Toyota, and BMW Brilliance are greater than 1.2. These three enterprises have good intrinsic growth, and their internal resources can fully support enterprises to obtain market competitive advantage and market share. The best intrinsic growth rate is 1.217 of Chang'an automobile. The intrinsic growth rate of Dongfeng Nissan is 0.296, which is the lowest among the sample enterprises in Table 4, which is also in line with the actual situation of Dongfeng Nissan's poor market performance in the past two years.

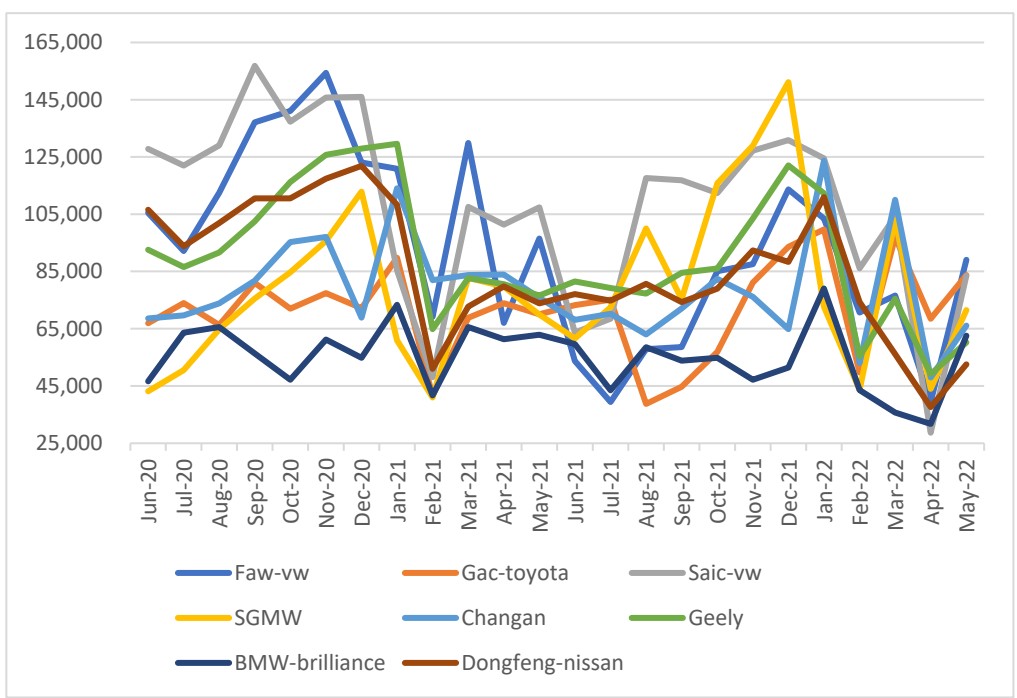

**Figure 3.** Change trend of sales data of top enterprises.

**Table 4.** Growth analysis of China's leading enterprises in fuel vehicle sales.

| Enterprise | Intrinsic Growth Rate ($\alpha1$) | Internal Inhibition Coefficient ($\gamma2$) | Theoretical Upper Limit of Sales Volume (K1) |
|---|---|---|---|
| Total national sales | 0.697 (2.758) *** | −0.000000372 (−2.909) *** | 1,878,113 |
| Faw-vw | 0.421 (1.615) * | −0.00000424 (−1.892) * | 99,518 |
| Gac-toyota | 1.214 (4.778) *** | −0.00001613 (−5.013) *** | 75,265 |
| SAIC-VW | 0.441 (1.303) | −0.00000405 (−1.499) * | 108,821 |
| SGMW | 0.620 (2.601) *** | −0.00000686 (−3.002) *** | 90,348 |
| Chang'an | 1.217 (5.843) *** | −0.00001462 (−6.330) *** | 83,228 |
| Geely | 0.410 (1.916) * | −0.00000439 (−2.136) ** | 93,256 |
| BMW-brilliance | 1.201 (5.188) *** | −0.00002081 (−5.412) *** | 57,710 |
| Dongfeng-nissan | 0.296 (1.385) | −0.00000352 (−1.619) * | 84,042 |

() *t* value, * *p* value < 0.1, ** *p* value < 0.05, *** *p* value < 0.01.

By comparing the data in Tables 2 and 4, it can be found that the intrinsic growth rate of other new energy vehicle enterprises except Nio automobile is significantly lower than the overall level of national automobile sales. The intrinsic growth rate of automobile sales of new energy automobile enterprises is far lower than that of high-quality enterprises in traditional automobiles. In order to better analyze the growth trend of automobile manufacturing enterprises, this paper further introduces the fuel vehicle enterprises that have significantly decreased sales and even face the risk of exiting the Chinese automobile market as a comparative sample.

As shown in Table 5, this paper selects four joint ventures with a significant decline in fuel vehicle sales as a comparative sample. These famous Sino foreign joint ventures include Chang'an Ford, SAIC Skoda, GAC Acura, and GAC jeep. The car sales of Chang'an Ford once ranked among the top 10 in China's auto market, and now the sales ranking hovers around 30. When observing the product strategy of Chang'an Ford, people were surprised to find that in a vigorous market, the new product update of Chang'an Ford had a strange "stagnation". The price of stagnant product update of Chang'an Ford is the decline of its sales volume and influence. At the same time, the decision-making mistakes have had a significant impact on the development of Chang'an Ford. The most typical

example is that Chang'an Ford used a three-cylinder engine on its main sales model, fox, which caused a precipitous decline in car sales. A similar situation occurred in several other enterprises with declining sales. GAC Acura and GAC Jeep have almost given up on the Chinese auto market. The change trend of the sample data of enterprises with declining auto sales in China is shown in Figure 4.

**Table 5.** Monthly sales volume of sample enterprises with declining automobile sales.

| Report Date | GAC Acura | Chang'an Ford | SAIC SKODA | GAC Jeep |
|---|---|---|---|---|
| May-22 | 617 | 16,296 | 3300 | 0 |
| Apr-22 | 458 | 9292 | 3284 | 1 |
| Mar-22 | 11 | 6931 | 1200 | 52 |
| Feb-22 | 412 | 13,031 | 5501 | 91 |
| Jan-22 | 36 | 8097 | 4270 | 132 |
| Dec-21 | 101 | 18,069 | 5800 | 1724 |
| Nov-21 | 5158 | 24,627 | 5704 | 1376 |
| Oct-21 | 376 | 25,412 | 7800 | 1829 |
| Sep-21 | 394 | 22,483 | 7601 | 2171 |
| Aug-21 | 445 | 22,930 | 6600 | 1735 |
| Jul-21 | 134 | 22,754 | 4400 | 724 |
| Jun-21 | 221 | 18,436 | 2900 | 528 |
| May-21 | 406 | 14,032 | 1900 | 555 |
| Apr-21 | 524 | 13,367 | 4000 | 1503 |
| Mar-21 | 643 | 10,074 | 8300 | 2176 |
| Feb-21 | 756 | 15,171 | 5000 | 2523 |
| Jan-21 | 362 | 8192 | 2500 | 2501 |
| Dec-20 | 825 | 22,331 | 5000 | 2502 |
| Nov-20 | 1224 | 25,661 | 7000 | 5176 |
| Oct-20 | 906 | 22,683 | 9000 | 3655 |
| Sep-20 | 1260 | 20,584 | 11,000 | 4007 |
| Aug-20 | 1163 | 21,388 | 13,500 | 3862 |
| Jul-20 | 802 | 15,740 | 16,000 | 3201 |
| Jun-20 | 1002 | 16,702 | 11,960 | 3034 |

As shown in Figure 4, the enterprises with declining sales volume listed in the sample have the following characteristics: the sales volume of some enterprises has remained at a very low level for a long time, such as GAC Acura and GAC jeep, which have actually withdrawn from the Chinese market. When Chang'an Ford was facing sales difficulties, through its own efforts, it achieved a certain increase in sales volume. Therefore, the sales trend of Chang'an Ford shows the characteristics of oscillation and fluctuation.

As shown in Table 6, the intrinsic growth rate of the sample enterprises with declining auto sales is not high. The intrinsic growth rate of Chang'an Ford is 0.417, which is the highest among the sample enterprises in Table 6. Chang'an Ford is also the best survivor of the Table 6 samples, with no signs of delisting or liquidation at this time. The situation of the other three enterprises is not optimistic as SAIC SKODA even has an extreme intrinsic growth rate of less than 0, which indicates that the company is actually no longer suitable to continue operating and should choose to withdraw from the Chinese market. GAC Acura and GAC Jeep also face the same choices as SAIC Skoda.

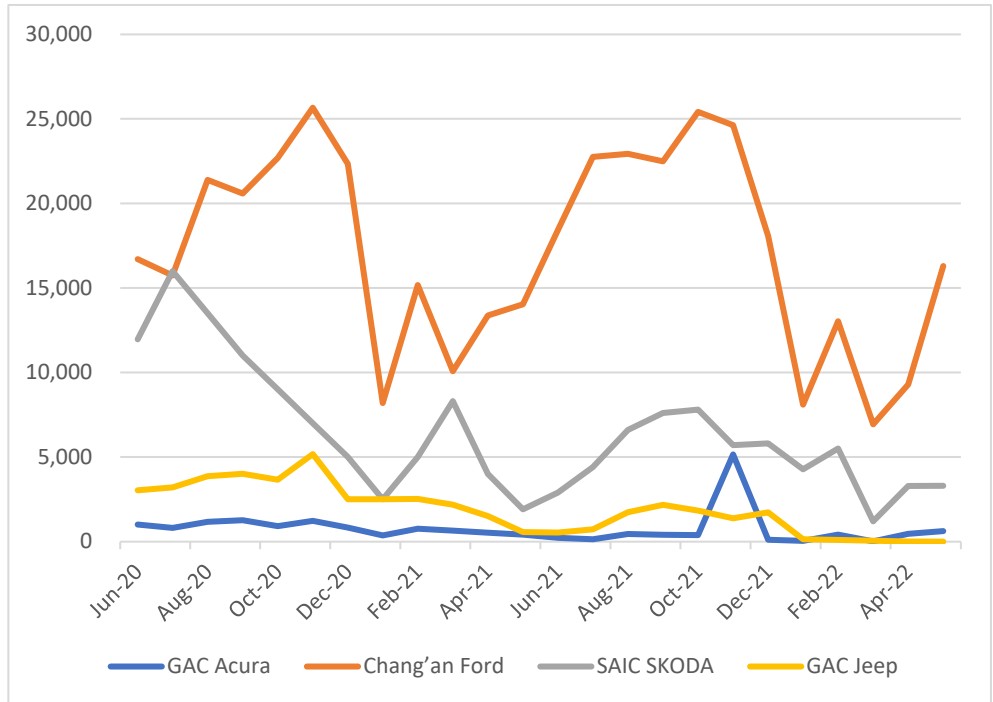

**Figure 4.** Change trend of sample data of enterprises with declining sales.

**Table 6.** Growth analysis of sample enterprises with declining auto sales.

| Enterprise | Intrinsic Growth Rate ($\alpha 1$) | Internal Inhibition Coefficient ($\gamma 2$) | Theoretical Upper Limit of Sales Volume (K1) |
|---|---|---|---|
| GAC Acura | 0.305 (0.803) | −0.00025116 (−2.868) *** | 1215 |
| Chang'an Ford | 0.417 (1.535) * | −0.00002241 (−1.743) * | 18,606 |
| SAIC SKODA | −0.018 (−0.106) | −0.00000867 (−0.588) | −2140 |
| GAC Jeep | 0.227 (1.080) | −0.00009241 (−1.583) * | 2462 |

() *t* value, * *p* value < 0.1, *** *p* value < 0.01.

### 3.2. Analysis of Market-Driven Mechanism Based on LV Model

In the natural ecological environment, the growth of a population cannot be separated from the support provided by the ecological environment, nor from the influence of other populations. Similarly, the growth of new energy vehicle enterprises depends on the development of the external economic environment, and the sales volume of new energy vehicles also depends on the overall demand of the automobile market. In the short term, the government's favorable policies may promote the increase of sales of new energy vehicles, and the sustainable development of new energy vehicles essentially depends on the benign development of the automobile market. This section uses the LV model to examine the impact of market capacity on new energy vehicles. This paper studies the market capacity expressed by the total national automobile sales in the same period.

As shown in Table 7, the vast majority of enterprises do not have significant market-driven mechanisms, and these companies are mainly exposed to the competitive mechanisms of the market.

**Table 7.** Analysis of market-driven mechanism of automobile sales.

| Enterprise | Intrinsic Growth Rate (α1) | Internal Inhibition Coefficient (γ2) | Theoretical Upper Limit of Sales Volume (K1) | γ3 | Symbiotic Influence Factor (β12) | Market Influence Mechanism |
|---|---|---|---|---|---|---|
| BYD | 0.425 (2.895) *** | −0.00000103 (−0.907) | 439,740 | −0.00000017 (−2.819) *** | −0.725 | Market competition |
| Tesla China | 0.481 (0.889) | −0.00001241 (−1.981) * | 38,757 | 0.00000005 (0.157) | 0.184 | Market driven |
| Xiaopeng | 0.707 (1.751) * | −0.00004523 (−1.994) * | 15,628 | −0.00000011 (−0.442) | −0.287 | Market competition |
| Hozon | 0.867 (3.674) *** | −0.00005261 (−2.882) *** | 16,494 | −0.00000020 (−1.681) * | −0.436 | Market competition |
| WM | 1.130 (2.559) * | −0.00010121 (−1.473) * | 11,161 | −0.00000042 (−1.762) * | −0.703 | Market competition |
| Lixiang | 0.738 (1.597) * | −0.00003884 (−1.359) | 19,004 | −0.00000019 (−0.687) | −0.491 | Market competition |
| Nio | 0.794 (2.198) ** | −0.00012581 (−3.784) *** | 6313 | 0.00000009 (0.495) | 0.218 | Market driven |
| Leap | 1.012 (2.226) ** | −0.00006453 (−1.956) * | 15,685 | −0.00000031 (−1.616) | −0.567 | Market competition |
| Faw-vw | 0.684 (1.626) * | −0.00000272 (−0.924) | 251,192 | −0.00000022 (−0.800) | −0.598 | Market competition |
| Gac-toyota | 1.109 (3.821) *** | −0.00001773 (−4.611) *** | 62,511 | 0.00000012 (0.778) | 0.203 | Market driven |
| SAIC-VW | 0.768 (1.844) * | −0.00000152 (−0.463) | 504,277 | −0.00000033 (−1.307) | −0.800 | Market competition |
| SGMW | 1.010 (2.267) ** | −0.00000463 (−1.476) * | 218,156 | −0.00000031 (−1.036) | −0.573 | Market competition |
| Chang'an | 0.767 (3.198) *** | −0.00001813 (−7.723) *** | 42,289 | 0.00000040 (2.844) *** | 0.971 | Market driven |
| Geely | 0.431 (1.522) * | −0.00000385 (−0.760) | 111,727 | −0.00000004 (−0.116) | −0.167 | Market competition |
| BMW-brilliance | 1.169 (4.025) *** | −0.00002113 (−4.923) *** | 55,332 | 0.00000003 (0.188) | 0.043 | Market driven |
| Dongfeng-nissan | 0.355 (1.345) * | −0.00000266 (−0.858) | 133,400 | −0.00000007 (−0.397) | −0.385 | Market competition |
| GAC Acura | −0.103 (−0.051) | −0.00027559 (−2.233) ** | −373 | 0.00000025 | −4.481 | Market competition |
| Chang'an Ford | 0.717 (1.906) * | −0.00001710 (−1.215) | 41,942 | −0.00000022 (−1.123) | −0.564 | Market competition |
| SAIC SKODA | 0.737 (1.442) | −0.00001256 (−0.770) | 58,654 | −0.00000038 (−1.614) * | −0.979 | Market competition |
| GAC Jeep | 0.303 (0.637) | −0.00008887 (−1.399) | 3405 | −0.00000005 (−0.187) | −0.283 | Market competition |

() *t* value, * *p* value < 0.1, ** *p* value < 0.05, *** *p* value < 0.01.

Based on the analysis results of LV model, when the influence coefficient of the total market volume on the sales volume of the enterprise is positive, it is considered that the new energy vehicles of the enterprise are facing the external market driven development mode. If the influence coefficient of the total sales volume of the automobile market on the

sales volume of the enterprise is negative, it is considered that the new energy vehicles of the enterprise are facing the external market competition mode.

Among the sample enterprises, only GAC Toyota, Chang'an, BMW Brilliance, Tesla China, and Nio are market driven. The development of the above five enterprises mainly depends on the development of China's automobile market. Among them, GAC Toyota, Chang'an, and BMW Brilliance are the dominant enterprises in traditional fuel vehicles, which is consistent with the previous analysis in this paper. Among the new energy vehicle enterprises, only Tesla China and Nio are market-driven enterprises. Nio has a small sales value, a small market share, and limited influence on the market. It can be seen that only Tesla China can achieve sustainable development under market-driven conditions. Of course, it is relatively one-sided to evaluate the development prospects of enterprises only from the two aspects of the intrinsic growth rate and market drive of enterprise growth. Therefore, this paper continues to use entropy weight TOPSIS method to carry out a more comprehensive evaluation of the growth prospects of enterprises. This study comprehensively considers the intrinsic growth rate given by logistic model and LV model, the theoretical upper limit of enterprise sales, the driving effect of market on enterprise sales and other indicators. The analysis results are shown in the table below.

In order to facilitate TOPSIS evaluation, the evaluation results of logistic and LV models are normalized as is shown in Table 8. In order to facilitate the calculation of entropy weight, a small value (0.001) is added to each evaluation value in the normalization process to ensure that all values are greater than 0. The evaluation results show that Chang'an, GAC Toyota, and BMW Brilliance ranked the top three in the evaluation of growth sustainability, which is consistent with the previous research conclusions. GAC Acura and GAC Jeep ranked last in terms of growth sustainability, which is also consistent with the previous research conclusion. The growth sustainability of new energy enterprises is generally not high, with BYD ranking 15th, Leap ranking 16th, and WM ranking 18th. Xiaopeng, Hozon, Nio, and Lixiang rank at the medium level in the ranking of the sample enterprises. Tesla China ranks fifth in terms of growth sustainability, which is the best comprehensive evaluation among new energy enterprises. The growth sustainability of most new energy vehicle enterprises needs to be improved. The main way to improve the growth sustainability is to change the growth mechanism of enterprises and change the enterprise development mode to market driven mode. This paper uses the LV MCGP model to optimize and simulate the market driven model and verify the market mechanism.

**Table 8.** Entropy weight TOPSIS evaluation results.

| Enterprise | Logistic Model | | | LV Model | | | | | TOPSIS Result | RANK |
|---|---|---|---|---|---|---|---|---|---|---|
| | $\alpha1$ | $\gamma2$ | K1 | $\alpha1$ | $\gamma2$ | K1 | $\gamma3$ | $\beta12$ | | |
| BYD | 0.0989 | 1.0001 | 1.0001 | 0.4152 | 1.0001 | 0.8722 | 0.3050 | 0.6890 | 0.6136 | 15 |
| Tesla China | 0.4600 | 0.9549 | 0.3083 | 0.4592 | 0.9587 | 0.0776 | 0.5733 | 0.8557 | 0.6833 | 5 |
| Xiaopeng | 0.4705 | 0.8002 | 0.0852 | 0.6369 | 0.8391 | 0.0318 | 0.3781 | 0.7694 | 0.6422 | 9 |
| Hozon | 0.4819 | 0.7559 | 0.0738 | 0.7627 | 0.8122 | 0.0335 | 0.2684 | 0.7420 | 0.6339 | 10 |
| WM | 0.4090 | 0.4170 | 0.0352 | 0.9694 | 0.6352 | 0.0230 | 0.0001 | 0.6931 | 0.5377 | 18 |
| Lixiang | 0.3961 | 0.8038 | 0.0746 | 0.6613 | 0.8624 | 0.0385 | 0.2806 | 0.7319 | 0.6232 | 14 |
| Nio | 0.7588 | 0.5248 | 0.0632 | 0.7053 | 0.5456 | 0.0133 | 0.6221 | 0.8620 | 0.6276 | 13 |
| Leap | 0.3305 | 0.7900 | 0.0609 | 0.8767 | 0.7688 | 0.0319 | 0.1342 | 0.7180 | 0.6084 | 16 |
| Faw-vw | 0.3556 | 0.9859 | 0.6537 | 0.6188 | 0.9939 | 0.4986 | 0.2440 | 0.7123 | 0.6614 | 6 |
| Gac-toyota | 0.9977 | 0.9384 | 0.4977 | 0.9529 | 0.9393 | 0.1247 | 0.6586 | 0.8592 | 0.8406 | 2 |
| SAIC-VW | 0.3718 | 0.9866 | 0.7135 | 0.6848 | 0.9983 | 1.0001 | 0.1099 | 0.6753 | 0.6533 | 8 |
| SGMW | 0.5167 | 0.9754 | 0.5947 | 0.8751 | 0.9870 | 0.4331 | 0.1342 | 0.7169 | 0.6930 | 4 |
| Chang'an | 1.0001 | 0.9444 | 0.5489 | 0.6841 | 0.9378 | 0.0846 | 1.0001 | 1.0001 | 0.8446 | 1 |
| Geely | 0.3467 | 0.9853 | 0.6134 | 0.4199 | 0.9898 | 0.2222 | 0.4635 | 0.7914 | 0.6608 | 7 |
| BMW-brilliance | 0.9871 | 0.9197 | 0.3849 | 1.0001 | 0.9269 | 0.1105 | 0.5489 | 0.8299 | 0.8074 | 3 |
| Dongfeng-nissan | 0.2544 | 0.9888 | 0.5542 | 0.3602 | 0.9942 | 0.2652 | 0.4269 | 0.7514 | 0.6300 | 12 |
| GAC Acura | 0.2616 | 0.0001 | 0.0217 | 0.0001 | 0.0001 | 0.0001 | 0.8172 | 0.0001 | 0.2268 | 20 |
| Chang'an Ford | 0.3523 | 0.9133 | 0.1335 | 0.6448 | 0.9416 | 0.0840 | 0.2440 | 0.7186 | 0.6336 | 11 |
| SAIC SKODA | 0.0001 | 0.9682 | 0.0001 | 0.6605 | 0.9581 | 0.1171 | 0.0489 | 0.6424 | 0.5699 | 17 |
| GAC Jeep | 0.1985 | 0.6339 | 0.0297 | 0.3193 | 0.6802 | 0.0076 | 0.4513 | 0.7701 | 0.5199 | 19 |
| wj | 0.130 | 0.164 | 0.073 | 0.156 | 0.167 | 0.022 | 0.119 | 0.169 | | |

As shown in Table 9, when the market drivers increased, the theoretical car sales of all enterprises increased significantly. Therefore, the market-driven model is a feasible development mechanism. Moreover, in this paper, the optimization value of LV MCGP model is completed under the condition that the total market capacity remains unchanged, which also shows that the sales volume growth and enterprise growth of new energy vehicles can be achieved while the total volume of China's automobile market remains unchanged.

**Table 9.** LV MCGP model optimization results.

| β12 | 0.10 | 0.25 | 0.40 | 0.55 | 0.70 | 0.85 | 1.00 |
|---|---|---|---|---|---|---|---|
| BYD | 154,134 | 161,413 | 168,693 | 175,972 | 183,251 | 190,530 | 197,809 |
| Tesla China | 36,344 | 38,664 | 40,985 | 43,306 | 45,627 | 47,947 | 50,268 |
| Xiaopeng | 22,635 | 23,709 | 24,783 | 25,857 | 26,931 | 28,004 | 29,078 |
| Hozon | 15,132 | 16,563 | 17,995 | 19,427 | 20,858 | 22,290 | 23,722 |
| WM | 5823 | 6332 | 6848 | 7350 | 7859 | 8368 | 8877 |
| Lixiang | 15,695 | 16,968 | 18,240 | 19,513 | 20,785 | 22,058 | 23,331 |
| Nio | 13,975 | 14,812 | 15,648 | 16,485 | 17,321 | 18,158 | 18,995 |
| Leap | 14,735 | 16,263 | 17,792 | 19,320 | 20,849 | 22,377 | 23,906 |

As shown in Table 10, in order to test the stability and sensitivity of parameter β and LV MCGP model, the range of β value is set from −2 to 2. Based on the theory of population dynamics, the range of β value should be between −1 and 1. Here, the value of β is beyond the theoretical range. The data in Table 10 show that even when the β value exceeds the theoretical range, the LV MCGP model can still be successfully optimized. The LV MCGP model has good model stability and parameter sensitivity.

**Table 10.** LV MCGP Model stability analysis.

| β12 | −2 | −1.2 | −1 | −0.6 | 0.6 | 1.2 | 2 |
|---|---|---|---|---|---|---|---|
| BYD | 52,226 | 91,048 | 100,754 | 120,165 | 178,398 | 207,515 | 246,337 |
| Tesla China | 3854 | 16,231 | 19,325 | 25,514 | 44,079 | 53,362 | 65,739 |
| Xiaopeng | 7603 | 13,330 | 14,761 | 17,625 | 26,215 | 30,510 | 36,236 |
| Hozon | <0 | 2723 | 4632 | 8450 | 19,904 | 25,631 | 33,267 |
| WM | <0 | 1411 | 2090 | 3447 | 7519 | 9555 | 12,270 |
| Lixiang | <0 | 4666 | 6362 | 9756 | 19,937 | 25,027 | 31,815 |
| Nio | <0 | <0 | 816 | 3047 | 9740 | 13,086 | 17,548 |
| Leap | <0 | 1488 | 3526 | 7602 | 19,830 | 25,944 | 34,096 |

## 4. Results and Discussion

This study analyzes the development of China's new energy vehicle manufacturing enterprises in detail from two aspects: the enterprise's own growth and the market-driven mode. The overall intrinsic growth rate of China's auto market remains at a relatively good level. The intrinsic growth rate of China's new energy vehicle manufacturing enterprises is generally lower than the overall level of China's automobile market, and also lower than high-quality fuel vehicle manufacturing enterprises. In terms of the mechanism of market driven enterprise growth, Chang'an, GAC Toyota, BMW Brilliance and other enterprises are at a relatively good level of coordination. The market-driven mechanism of the vast majority of enterprises is not significant, and these enterprises are mainly facing the market competition mechanism. Among the sample enterprises, only GAC Toyota, Chang'an, BMW Brilliance, Tesla China, and Nio are market-driven. Among the new energy vehicle enterprises, only Tesla China and Nio are market-driven enterprises.

This paper uses entropy weight TOPSIS method to make a more comprehensive evaluation of the growth prospects of enterprises. The evaluation results show that the growth sustainability of Chinese new energy enterprises is generally not high, with BYD ranking 15th, Leap ranking 16th, and WM ranking 18th. Xiaopeng, Hozon, Nio, and Lixiang rank at the medium level in the ranking of the sample enterprises. Tesla China ranks fifth in terms of growth sustainability, which is the best comprehensive evaluation among new energy enterprises. The growth sustainability of most new energy vehicle enterprises needs to be improved, and these enterprises need to change their development mode to market-driven mode.

The existing research on new energy vehicles in China mainly focuses on industrial development [1] and the impact of government policies such as subsidies [2–5]. Compared with the above research, this paper focuses on the micro development of new energy vehicle enterprises and expands the research boundary and perspective. Industrial characteristics should not replace the development of enterprise personality. The dynamic nature of government policy will also lead to poor timeliness of policy research. This paper takes the new energy vehicle enterprises as the starting point and establishes a cross-level research mechanism. There are not many existing studies on new energy enterprises, such as the research on technological innovation efficiency of enterprises [6]. Compared with the research on technological innovation efficiency, this paper has no data and mechanism limited to the enterprise level. This paper constructs a research model of market symbiosis mechanism.

Compared with existing research methods [16–19], this paper uses a sustainable development evaluation method based on population ecological model, which opens up a new field of quantitative analysis. The advantage of the evaluation method proposed in this paper is that it cannot only rank the development sustainability of new energy vehicle manufacturing enterprises, but also analyze the development dynamic mechanism of new energy vehicle manufacturing enterprises. At the same time, the method proposed in this paper can present the dynamic characteristics of enterprise development, which has advantages over the traditional comprehensive evaluation method based on the evaluation index system. This paper focuses on the sustainability of the development of new energy vehicle manufacturing enterprises from a new perspective. Relevant literature believes that China's new energy vehicle industry has changed from "government-driven" to "government-driven + market-driven" [25]. This study does not support this argument; instead, this paper finds that most Chinese new energy manufacturing enterprises have not realized the "market-driven" development model.

## 5. Conclusions

This paper takes the development sustainability evaluation of China's new energy vehicle manufacturing enterprises as the research goal. In order to clarify this research topic, this paper constructs a set of enterprise growth sustainability evaluation methods based on population dynamics model. The empirical results show that the logistic and LV models in this paper are effective analytical methods.

The highlights of this paper are as follows. (1) A logistic model is used to analyze the intrinsic growth and internal inhibition coefficient of enterprise growth. (2) The market-driven mode of China's new energy enterprises is analyzed using the LV model. (3) The growth sustainability of China's new energy enterprises is comprehensively evaluated using the entropy weight TOPSIS method. (4) The LV-MCGP model is constructed to optimize and simulate the enterprise development mode.

The defects of this paper lie in the following aspects. (1) This paper does not specifically analyze the life cycle of the sample enterprises, and the methodology proposed in this paper does not specifically analyze the specific stage of the sample enterprises in their life cycle. (2) The ecosystem on which the survival of enterprise population depends is complex, which is not reflected in this paper. Future research can be carried out from the following aspects: (1) The development and sustainability of enterprises in combination with life

cycle theory can be analyzed. (2) The symbiotic relationship of enterprise population from the perspective of ecosystem can be analyzed.

**Funding:** This research was funded by Jiangsu social science application research project, grant number 22SYB-089. And the APC was funded by 22SYB-089.

**Institutional Review Board Statement:** Not applicable.

**Informed Consent Statement:** Not applicable.

**Data Availability Statement:** All relevant data have been submitted within this manuscript.

**Conflicts of Interest:** The author declares no conflict of interest.

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
