# Peer review of "Exploring the Sustainability of China’s New Energy Vehicle Development: Fresh Evidence from Population Symbiosis"

_sustainability, doi:10.3390/su141710796_

Round 1

Reviewer 1 Report

1. The literature review needs to be supplemented. Lines 124-135 of the manuscript give the marginal academic contribution of this paper. Before that, the author should focus on the innovation of the article to conduct systematic literature review and comparative analysis, and use this as a foundation to highlight the academic innovation of the manuscript.

2. In section 2.1 of the manuscript, the author discusses new energy vehicles. The background knowledge introduction here does not need to be introduced too much. It is not recommended for the author to put it in the methodology and data section. It can be considered to be simplified and placed in the introduction.

3. Lines 176-246 of the manuscript need to be refined. It is recommended that the author outline the core ideas of the theoretical method and its adaptability and scientificity. The rest of the content can be considered in the introduction and used as a foreshadowing to highlight the innovative points of the article.

4. The author extends the Logistic and LV models to the research on the growth mechanism of NEV, and incorporates the model estimation parameters into the TOPSIS evaluation system, and then uses the MCGP model to analyze the optimization path of the growth model of new energy vehicle companies. It is recommended that the author illustrate the research ideas in a graphical way, which is easier for readers to understand. In addition, in lines 283-289 of the manuscript, I suggest that the author make clear the parameters that need to be empowered, and explain the basis for their selection, which is directly related to the rationality of the selection of new energy vehicle growth sustainability indicators.

5. Do the entropy weight method and TOPSIS model draw on previous research by scholars, or are the authors themselves proposed? This issue needs to be clarified, and it is recommended that the author support it with literature. Such as Zhao J., et al. Environmental vulnerability assessment for mainland China based on entropy method. Ecol. Ind. 2018, 91:410-422; Wang D., et al. Evaluating urban ecological civilization and its obstacle factors based on integrated model of PSR-EVW-TOPSIS: A case study of 13 cities in Jiangsu Province, China. Ecol. Ind. 2021, 133:108431.

6. Lines 400-539 of the manuscript are proposed as a separate chapter "Empirical Analysis".

7. In the part of empirical analysis and result discussion, I suggest that the author compare the findings of this paper with other relevant research conclusions, and it is necessary to supplement some literature to support the viewpoints of this paper.

8. Can you explain the connection and difference between market driving and market competition in Table 7?

9. The formula typesetting does not meet the requirements of the journal and needs to be corrected.

10. The language presentation of the manuscript needs to be improved.

Author Response

1. The author supplemented 15 articles and adjusted the literature review. 2. The author simplified the corresponding contents and put them in the introduction. 3. The author adjusted the relevant statements. 4. The author defined the relevant parameters. 5. The combination of these two methods is an attempt made by the author in the previous literature, and a new reference is added in this paper. 6. Lines 400-539 of the manuscript are proposed as a separate chapter "empirical analysis". 7. Supplementary discussion 8. Supplementary description 9. The format has been modified. The modified parts are marked in red font

Reviewer 2 Report

·       The two paragraphs of the introduction section lack references.

·       Line 85-87 Page 2: the statement should be considered more before putting forward. Is it only the PV industry regarded as the foundation? How about the other renewable energy industries such as wind?

·       There is a large gap in the literature review. Please look into the recent scientific research before presenting the contribution of this paper.

·       Section 2.1 New Energy Vehicles is not belonging to any methodology and data. Maybe move this section to a separate section.

·       Line 346 and 360 Page 8/ Line 383 Page 9: the data source should be cited as reference. Please do not simply copy the webpage in the main text.

·       It would be better to make table 1 into a figure to show the trend. Same for table 3 and 5.

·       Section 2.4 (1) Growth analysis of single population based on logistic model is not a part of the methodology, right? Why not move it to the results section?

·       Line 562 Page 17: a literature review is needed to support your statement. To be honest, the methodology adopted in this work is quite simple.

·       It is wired to include highlights in the conclusion section. Please make the highlights a separate part.

·       What is the main conclusion of this paper? What can we learn from this work?

Author Response

1. References are supplemented. 2 the author added explanatory text. 3 the author supplemented the literature review. 4 the author made modifications. 5 the author made modifications. 6 the author supplemented the charts. 7 the author made modifications. 8 the author added notes. 9 the author made modifications. The modified parts are marked in red font

Reviewer 3 Report

The knowledge gap must be explained carefully, and the paper's contribution should be defined to fill the mentioned knowledge gap. 

It is suggested to add a table to summarize the survey outcome.

Managing the charging demand of electric vehicles through smart charging is a crucial aspect of expanding the use of electric vehicles in transportation fleets. In order to accurately anticipate the daily demand for EVs, reliable and precise data-driven methodologies are required. The literature survey should be enriched with more recently published studies in EV smart charging based on novel data-driven methods, such as a novel cross-case electric vehicle demand modeling based on 3d convolutional generative adversarial networks, plug-in electric vehicle behavior modeling in energy market: a novel deep learning-based approach with clustering technique, and charging demand of plug-in electric vehicles: forecasting travel behavior based on a novel rough artificial neural network approach.

More technical details about Sections 2.2 and 2.3 should be added.

The overall structure of the proposed method is too general, and a Pseudo-code is needed to explain the details. 

The discussion part (Section 3) should be enriched with more informative details about the findings of this study, for instance, the reviewer cannot follow the authors’ ideas behind some tables.

A brief comparison with related works is also suggested in Section 3. 

The conclusion should concentrate more on the results.

Author Response

The knowledge gap must be explained carefully, and the paper's contribution should be defined to fill the mentioned knowledge gap. It is suggested to add a table to summarize the survey outcome.

Response 1: the author added the literature and revised the literature review.

Managing the charging demand of electric vehicles through smart charging is a crucial aspect of expanding the use of electric vehicles in transportation fleets. In order to accurately anticipate the daily demand for EVs, reliable and precise data-driven methodologies are required.

Response 2: the author added relevant literatures and citations.

The literature survey should be enriched with more recently published studies in EV smart charging based on novel data-driven methods, such as a novel cross-case electric vehicle demand modeling based on 3d convolutional generative adversarial networks, plug-in electric vehicle behavior modeling in energy market: a novel deep learning-based approach with clustering technique, and charging demand of plug-in electric vehicles: forecasting travel behavior based on a novel rough artificial neural network approach.

Response 3:the author added relevant instructions

More technical details about Sections 2.2 and 2.3 should be added. The overall structure of the proposed method is too general, and a Pseudo-code is needed to explain the details. 

Response 4:The author added relevant explanations

The discussion part (Section 3) should be enriched with more informative details about the findings of this study, for instance, the reviewer cannot follow the authors’ ideas behind some tables.A brief comparison with related works is also suggested in Section 3. The conclusion should concentrate more on the results.

Response 5:the author added the content of discussion.

Round 2

Reviewer 1 Report

The author has refined the question I asked. I recommend accepting this article.

Reviewer 3 Report

Comments are addressed carefully.